# Early Response of Soil Properties under Different Restoration Strategies in Tropical Hotspot

Katarzyna A. Koryś [1,2,*,†], Agnieszka E. Latawiec [1,2,3,4,†], Maiara S. Mendes [1,2], Jerônimo B. B. Sansevero [5], Aline F. Rodrigues [1,2], Alvaro S. Iribarrem [1,2], Viviane Dib [1,2,6], Catarina C. Jakovac [7], Adriana Allek [2,6], Ingrid A. B. Pena [1,2], Eric Lino [1,2] and Bernardo B. N. Strassburg [1,2,7]

1    International Institute for Sustainability, R. Dona Castorina 124, Rio de Janeiro 22460-320, Brazil; a.latawiec@iis-rio.org (A.E.L.); m.mendes@iis-rio.org (M.S.M.); a.rodrigues@iis-rio.org (A.F.R.); a.iribarrem@iis-rio.org (A.S.I.); v.dib@iis-rio.org (V.D.); ingrid.pena@aluno.puc-rio.br (I.A.B.P.); e.lino@iis-rio.org (E.L.); b.strassburg@iis-rio.org (B.B.N.S.)
2    Rio Conservation and Sustainability Science Centre, Department of Geography and the Environment, Pontifical Catholic University of Rio de Janeiro, R. Marquês de São Vicente, 225-Gávea, Rio de Janeiro 22451-000, Brazil; adriana.allek@ufrj.br
3    Department of Production Engineering, Logistics and Applied Computer Science, Faculty of Production and Power Engineering, University of Agriculture in Kraków, Balicka 116B, 30-149 Kraków, Poland
4    School of Environmental Sciences, University of East Anglia, Norwich Research Park, Norwich NR4 7TJ, UK
5    Departament of Environmental Science, Federal Rural University of Rio de Janeiro (UFRRJ), Rodovia BR 465, km 07, Seropédica, Rio de Janeiro 23890-000, Brazil; sansevero@ufrrj.br
6    Programa de Pós-Graduação em Ecologia, Federal University of Rio de Janeiro, 68020, Rio de Janeiro 21941-901, Brazil
7    Forest Ecology and Management Group, Wageningen University, 6700 AA Wageningen, The Netherlands; catarina.jakovac@wur.nl
*    Correspondence: k.korys@iis-rio.org
†    These authors contributed equally to this article.

**Abstract:** The Brazilian Atlantic Forest has undergone adverse land-use change due to deforestation for urbanization and agriculture. Numerous restoration initiatives have been taken to restore its ecosystem services. Deforested areas have been restored through active intervention or natural regeneration. Understanding the impact of those different reforestation approaches on soil quality should provide important scientific and practical conclusions on increasing forest cover in the Brazilian Atlantic Forest biome. However, studies evaluating active planting versus natural regeneration in terms of soil recovery are scarce. We evaluate soil dynamics under those two contrasting strategies at an early stage (<10 years). Reforestation was conducted simultaneously on degraded lands previously used for cattle grazing and compared to an abandoned pasture as a reference system. We examined soil physicochemical properties such as: pH, soil organic matter content, soil moisture, N, P, K, Ca, Mg, Na, Fe, Mn, Cu, Al, and soil texture. We also present the costs of both methods. We found significant differences in restored areas regarding pH, Na, Fe, Mn content, and the cost. Soil moisture was significantly higher in pasture. Our research can contribute to better decision-making about which restoration strategy to adopt to maximize restoration success regarding soil quality and ecosystem services in the tropics.

**Keywords:** environmental decision-making; forest restoration; restoration strategy; soil recovery; tropical soils

## 1. Introduction

Large-scale deforestation has been recognized as a major environmental problem worldwide [1,2]. Legislative negligence in the enforcement of existing legislation has contributed to forest loss, spurring conversion of forested land for agricultural, economic, transport, or urban purposes [3]. Restoration is now an international priority to recuperate

degraded ecosystems and their services [2,4]. One of the crucial areas for restoration in Brazil is the Atlantic Forest.

The Brazilian Atlantic Forest is one of the most valuable biomes in Brazil. The biome has been recognized as a tropical biodiversity hotspot that includes unique endemic species and provides local and global ecosystem services such as climate regulation, water, and nutrient cycling [5,6]. Originally, covering about 150 million ha [7], it went through a period of deforestation [8] as the result of land-use change for urban development and agricultural production, most recently, for cattle ranching [9]. Approximately 12% of the original vegetation cover remains [10] (Supplementary Materials, Figure S1), and over than 90% of the remaining area is in private lands [11,12]. Deforestation is responsible for loss of soil nutrients, soil erosion, and biodiversity loss [13,14].

The beneficial effect of forest regrowth on biodiversity and soil organic carbon dynamics has been well documented [15–18]. For example, the meta-analysis conducted by Crouzeilles et al. [17] found that the success of restoring biodiversity and vegetation structure was, respectively, 34–56% and 19–56% higher in natural regeneration than in active restoration systems. In turn, De Medeiros et al. [18] reported a 20% gain in soil organic carbon due to natural regeneration in areas previously used for intensive farming systems in Brazil. However, changes in soil properties and quality due to afforestation are still poorly studied and often overlooked in restoration projects [18–20]. A systematic review by Mendes et al. [19] showed that the majority (59%) of the studies on restoration in Brazilian Atlantic Forest did not include any soil quality indicator. This result points to a soil data gap in restoration projects in the Brazilian Atlantic Forest biome. Closing this gap is critical to validating the most effective restoration strategy. Considering and demonstrating recovery of soil quality in the short term are important with regard to communication with landowners about more sustainable land management and forest restoration on their lands [20].

Soil is an integral part of the forest ecosystem and plays a key role in successful forest regeneration [21,22]. As soil properties may affect seedlings' establishment, growth, and survival [23]. Intrinsic interactions between soil and vegetation maintain an optimal nutrient cycle and forest dynamics [24]. Furthermore, diverse tree species may have different impacts on soil fertility, and consequently contribute differently to forest growth [25–27]. At a regional scale, land-use change may contribute rapidly to diminish soil quality [28]. Numerous studies have shown a negative effect of deforestation on soil organic matter, nitrogen, and soil microbiota activity, which converts soil to a low fertility state and results in lower productivity [29,30]. Enhancing and maintaining soil quality are necessary to guarantee environmental sustainability and forest recovery [31–33].

Active restoration and natural regeneration are two important strategies for restoring large areas of degraded and deforested tropical lands. Active restoration consists of management techniques such as planting seeds or seedlings. This technique requires often soil amendment and correction [34]. Natural regeneration occurs when the factors that cause environmental stress (e.g., cattle grazing) are removed and secondary succession can proceed, and generally no active management of the soil or vegetation is undertaken. Deciding which strategy should be applied depends on many factors, such a level of land degradation, natural rate of ecosystem recovery, financial resources, and the final objective of the restoration project [34,35]. In this study, we evaluate and compare soil properties under two restoration strategies: active and natural regeneration in the early stage (<10 years) of recovery, in tropical conditions. We address three questions: (i) what changes occur in soils in early stages of forest restoration? (ii) how does the restoration strategy affect soil properties and soil ecosystem services? and (iii) which restoration method is more effective for the soil recovery in the short term? Our research aims to increase the knowledge of tropical soils and their response to forest regrowth, to establish the most successful strategy on abandoned lands previously used for cattle grazing. The results obtained here may be useful to monitor occurrences belowground, monitor the

effectiveness of the restoration approach in terms of soil ecosystem services recovery, and provide insights for further research.

## 2. Materials and Methods

### 2.1. Study Site

The study was conducted at Fazenda Dourada ("Golden Farm", in English), located in the municipality of Casimiro de Abreu, Rio de Janeiro State (22°44′17.41″ S; 42°07′19.25″ W inside the Poço das Antas biological reserve insouth-eastern Brazil (Figure 1). Fazenda Dourada, formerly privately owned, was acquired by the Golden Lion Tamarin Association (Associação do Mico Leão Dourado, AMLD, in Portuguese). AMLD, in collaboration with local communities, city councils and landowners, aims to restore part of the Brazilian Atlantic Forest by creating forest corridors in the region that allow the free movement of native species and enhance biodiversity conservation [36]. This area is characterized by a high degree of perturbation and fragmentation; however, it shows a great diversity and richness of forest fragments since many endemic species occur in it. The region plays, inter alia, a key role in the protection of the habitat of the endangered golden lion primate tamarin (*Leontopithecus rosalia*), an endemic species of the Brazilian Atlantic Forest [37,38]. The predominant vegetation is Dense Ombrophilous Forest in different successional stages [39]. The climate is As (tropical rainy with dry season in winter) according to Köppen's classification, and mean annual temperature and precipitation are 25.5 °C and 1500–2000 mm, respectively [40]. The soils in the study area are Ultisols and fulvic Neosol [41]. These soils correspond to Acrisols and Fluvisols according to the FAO classification [42] and are characterized by low P availability and high $Al^{3+}$ content [43,44].

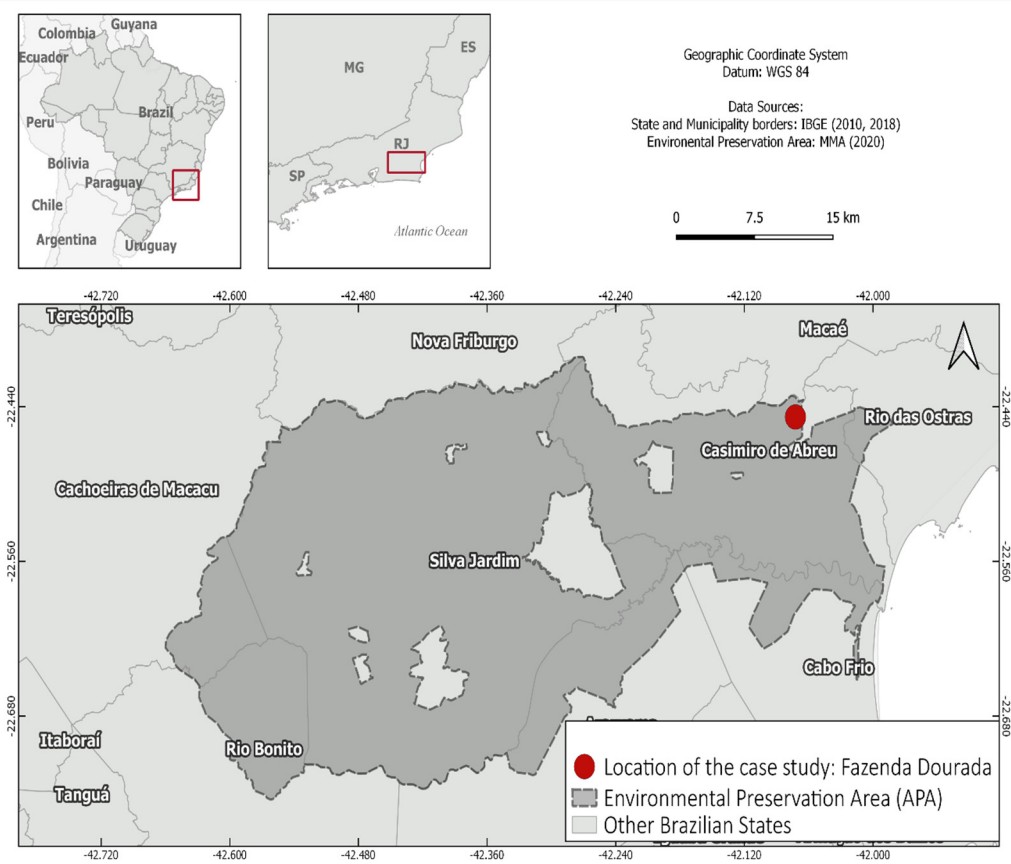

**Figure 1.** Location of the Fazenda Dourada, Rio de Janeiro State, Brazil. Source: International Institute for Sustainability. (Map of the case study was generated by software QGIS 3.4 "MADEIRA").

### 2.2. Experimental Design

Both evaluated strategies were implemented simultaneously, and the time elapsed since restoration began was approximately seven years. Restored areas were previously used as pastures for cattle grazing. Active restoration (planting) was established on three sites: two on the slope and one at the flat area on a riverbank. Tree-planting was conducted irrespectively by two institutions: AMLD (P1, P3) and The State Environment Institute (Instituto Estadual do Ambiente, Inea, in Portuguese) (P2). Natural regeneration (REG) occurred on the flat area. This area was isolated and monitored by AMLD. Site preparation was performed through manual weeding and mowing followed by cover fertilization (NPK-20-05-20) for all tree planting sites. All restored areas had been fenced. As a reference system we used abandoned pasture located nearby the restored sites (also refers to "control site", CON).

Restoration strategies were as follows:

P1: active, conducted on the slope.

P2: active, conducted on the slope.

P3: active, conducted on the flat area.

REG: natural regeneration occurring on the flat area.

Reference system/control site:

CON: abandoned pasture located on a flat ground, previously used for cattle grazing. Currently overgrown by grass (*Paspalum mandiocanum*). No evidence of cattle grazing was observed during the sampling: no signs of animal prints or excrement. The exposed fragments of soil indicated erosion processes.

For sampling, three blocks of 30 m × 60 m were designated at each planting site (P1, P2, and P3) and three blocks of 30 m × 60 m at natural regeneration (REG) site. Nine plots were randomly installed in each block, for a total of twenty-seven plots per restored site. We sampled approximately 1.07 ha per restored area.

### 2.3. Soil Sampling and Analysis

Soil sampling was carried out twice: in March/2016 and November/2017. Ten topsoil samples (0–20 cm) were collected from each plot with an auger and combined into one composite sample (n = 27 per restored site). On the control site, ten soil samples were collected randomly at a depth of 20 cm (Supplementary Materials Tables S1 and S2). Soil samples were homogenized and sieved to 2 mm and analyzed for pH ($H_2O$), soil moisture content (%); total N (g $kg^{-1}$); total P (mg $dm^{-3}$); residual P (mg $L^{-1}$); total K (mg $dm^{-3}$); total Na (mg $dm^{-3}$); total Mg (cmol $dm^{-3}$); Ca (cmol $dm^{-3}$); Al (cmol $dm^{-3}$); H + Al (cmol $dm^{-3}$); Fe (mg $dm^{-3}$); Mn (mg $dm^{-3}$), and Cu (mg/$dm^3$); soil organic matter (dag $kg^{-1}$); cation exchange capacity (CEC) (effective and potential; cmol $dm^{-3}$); and soil texture: clay (dag $kg^{-1}$), silt (dag $kg^{-1}$), and sand (dag $kg^{-1}$). Nitrogen was determined using the Kjeldahl method. Phosphorus (P), potassium (K), and sodium (Na) were analyzed using a Mehlich 1 extractor (0.05 mol/L HCl and 0.0125 mol/L $H_2SO_4$). Magnesium (Mg) and calcium (Ca) were extracted with 1 mol/L KCl solution. Iron (Fe) manganese (Mn) and copper (Cu) were analyzed using a Mehlich-1 extractor (in relation to soil extractor 1:10). To determine soil organic matter (SOM) content, $Na_2Cr_2O_7$ + $H_2SO_4$ 10N oxidation was applied [45]. Carbon content (C) was estimated using the van Bemmelen factor (=1.724) [46], and then the C: N ratio was calculated (Supplementary Materials Table S3). The potential CEC was measured as the sum of the base cations $Ca^{2+}$, $Mg^{2+}$, $Na^+$, and $K^+$ in addition to $Al^{3+}$ and $H^+$ (cmol kg–b 1). The effective CEC was defined as the sum of base cations in addition to $Al^{3+}$ (determined using a 1 mol/L KCl solution). Analysis of the soil moisture content (SMC) was based on the methodology described by Klute [47], and for the analysis of the soil texture we used the sedimentation fractionation methodology described by Gee and Bauder [48].

## 2.4. Characteristic of the Aboveground Vegetation

Vegetation parameters such as vegetation structure and density, species richness, diversity, and the floristic composition from restored areas have been described in detail by Galvão [49]. In summary: a total number of 960 individuals were sampled, and 63 species were found distributed in 26 families. The *Fabaceae* family had the highest species richness, with 21 species identified, followed by the *Euphorbiaceae* family with 4 species identified. The study shows that the site classified as P1 presents the highest species richness, with 37 species in total and 11 regenerating species, followed by P3 with a total of 34 identified species, 14 of these regenerating individuals, and P2 encompassing only 2 regenerating species with a richness of 29 species. P2 shows the highest number of individuals per hectare, approximately 1074, followed by P1, P3, and REG with the lowest density value— 736 individuals per hectare. Additionally, the site where natural regeneration occurred presents the lowest species richness (24 species sampled in total) [49]. The results for the vegetation are summarized in Table 1. A complete list of plant species (trees and shrubs) from each restored site can be found in Supplementary Materials Table S4.

**Table 1.** Characteristic of the vegetation cover on restored areas, Fazenda Dourada, Brazil. P1: active/slop; P2: active/slop; P3: active/flat; and REG: natural regeneration, adapted from Galvão [49].

| Restored Site: | Density (Number of Individuals/Ha) | Richness | Shannon Diversity Index (H') |
|:---:|:---:|:---:|:---:|
| P1: | 944 | 37 | 3.13 |
| P2: | 1074 | 29 | 3.00 |
| P3: | 855 | 34 | 2.99 |
| REG: | 736 | 24 | 2.43 |

## 2.5. Statistical Analyses

Statistical analyses were performed in R software version 3.4.4 [50]. We computed mean values and the standard deviation of each set of soil variables that was measured in each study site and the corresponding values for the reference system/control. We report the values of each variable as the difference between their mean value and the mean value of the control site (CON). Uncertainties for these differences were obtained through quadratic propagation of standard deviations [51]. To present results, we also calculate variation of a given soil property measurement. Graphical results for selected soil properties: Na, Fe, Mn, pH, SMC, clay, silt, sand, N, P, K, C: N, and SOM are presented in Supplementary Materials Figures S2–S9, Word file. For instance, for pH we present a difference between the mean value of pH measurements in the restored area (P1, P2, P3, and REG) and the mean value of that same soil property measured in an abandoned pasture (control site). The description of the quadratic propagation of standard deviations method is presented in Supplementary Materials Word file.

## 3. Results

### 3.1. Soil Properties

Our results indicate no significant difference in soil properties between restored areas, except for Na and pH (Table 2). Soil from pasture (CON) has the highest Na content ($25.57 \pm 1.90$), followed by the natural regeneration area (REG) ($22.44 \pm 3.70$) and P3 area ($22.44 \pm 5.18$) Only P2 showed significant difference ($p < 0.05$) of Na in relation to both other treatments and CON. Regarding soil pH, the value from active restoration in area P3 was significantly lower ($4.50 \pm 0.35$) from P1 ($5.50 \pm 0.28$) and P2 ($5.50 \pm 0.20$) areas. Soil moisture content (SMC), Fe, and Mn significantly differed in restored areas with respect to pasture. The value of soil moisture content in CON was higher ($27.25 \pm 7.70$) than in the restored areas. The lowest value of SMC was in areas P1 ($2.02 \pm 0.26$), P2 ($2.10 \pm 0.53$), P3 ($2.57 \pm 1.18$), and REG ($2.74 \pm 1.74$), respectively. There was no significant difference in SMC between the restoration strategies. The highest values of Fe were observed on

pasture (465.48 $\pm$ 143.19), followed by area P3 (255.63 $\pm$ 112.59) and natural regeneration area (REG) (162.61 $\pm$130.59). In the P1 and P2 plantation areas, the values of Fe presented a significant difference in relation to CON and the lowest value of Fe found in area P2 (27.10. $\pm$ 7.67). The Mn content varied between 28.99 $\pm$ 5.31 to 161.65 $\pm$ 53.63 mg dm$^{-3}$. The highest values of Mn were in the area P2 and P1 (161.65 $\pm$ 53.64 and 125.60 $\pm$ 37.19, respectively), in addition to the areas that showed significant difference ($p < 0.05$) of Mn in relation to CON. Lowest Mn value was observed in CON (28.99 $\pm$5.31). In case of soil organic matter (SOM), C:N ratio and the contents of macronutrients such as total N, P, and K, no significant differences were found between the restored areas and in relation to CON, regardless of the restoration strategy. The total N values ranged from 2.07 $\pm$ 0.37 to 2.43 $\pm$ 0.41. The lowest value was found in CON (2.07 $\pm$ 0.38), and the highest value was found on the area REG: (2.43 $\pm$ 0.42). In case of soil P, the lowest value was reported for CON (1.13 $\pm$ 0.94) and the highest was found in the active restoration area on the P1 slope with (3.21 $\pm$ 2.47). The K content ranged from 59.0 $\pm$ 36.20 to 114.39 $\pm$ 37.12. The highest K value was found in area P1 (114.39 $\pm$ 37.12) and the lowest in CON (61.54 $\pm$ 68.91).

**Table 2.** Mean values and $\pm$SD of soil properties (0–20 cm) of each restored area and degraded pasture (control), Fazenda Dourada, Brazil.

| Soil Property | P1 | P2 | P3 | REG | CON |
|---|---|---|---|---|---|
| pH (H$_2$O) | 5.50 $\pm$ 0.28 | 5.50 $\pm$ 0.20 | 4.50 $\pm$ 0.35 * | 4.75 $\pm$ 0.48 | 5.35 $\pm$ 0.39 |
| N-total (g kg$^{-1}$) | 2.22 $\pm$ 0.59 | 2.24 $\pm$ 0.49 | 2.16 $\pm$ 0.40 | 2.43 $\pm$ 0.42 | 2.07 $\pm$ 0.38 |
| K (mg dm$^{-3}$) | 114.39 $\pm$ 37.12 | 66.04 $\pm$ 34.20 | 60.03 $\pm$ 35.02 | 59.0 $\pm$ 36.20 | 61.54 $\pm$ 68.91 |
| P (mg dm$^{-3}$) | 3.21 $\pm$ 2.47 | 2.22 $\pm$ 0.36 | 2.92 $\pm$ 0.63 | 2.76 $\pm$ 1.09 | 1.13 $\pm$ 0.94 |
| Na (mg dm$^{-3}$) | 20.58 $\pm$ 1.95 | 11.26 $\pm$ 3.29 * | 22.44 $\pm$ 5.18 | 22.44 $\pm$ 3.70 | 25.57 $\pm$ 1.90 |
| SOM (dag kg$^{-1}$) | 1.83 $\pm$ 0.32 | 1.76 $\pm$ 0.24 | 1.24 $\pm$ 0.25 | 1.46 $\pm$ 0.30 | 1.77 $\pm$ 0.50 |
| Ca (cmol$_c$ dm$^{-3}$) | 1.77 $\pm$ 0.56 | 1.0 $\pm$ 0.29 | 0.67 $\pm$ 0.22 | 0.88 $\pm$ 1.00 | 0.95 $\pm$ 0.51 |
| Mg (cmol$_c$ dm$^{-3}$) | 0.87 $\pm$ 0.22 | 0.55 $\pm$ 0.14 | 0.53 $\pm$ 0.22 | 0.61 $\pm$ 0.37 | 0.45 $\pm$ 0.24 |
| Al (cmol$_c$ dm$^{-3}$) | 0.23 $\pm$ 0.12 | 0.20 $\pm$ 0.20 | 1.12 $\pm$ 0.41 | 0.99 $\pm$ 0.50 | 0.73 $\pm$ 0.31 |
| SMC (%) | 2.02 $\pm$ 0.26 ** | 2.10 $\pm$ 0.53 ** | 2.57 $\pm$ 1.18 ** | 2.74 $\pm$1.74 ** | 27.25 $\pm$ 7.70 |
| Fe (mg dm$^{-3}$) | 76.79 $\pm$ 24.87 ** | 27.10 $\pm$ 7.67 ** | 255.63 $\pm$ 112.59 | 162.61 $\pm$ 130.59 | 465.48 $\pm$ 143.19 |
| Mn (mg dm$^{-3}$) | 125.60 $\pm$ 37.19 ** | 161.65 $\pm$ 53.64 ** | 51.17 $\pm$ 38.10 | 70.30 $\pm$ 44.83 | 28.99 $\pm$ 5.31 |
| Cu (mg dm$^{-3}$) | 1.66 $\pm$0.47 | 1.42 $\pm$ 0.31 | 2.51 $\pm$ 0.01 | 1.08 $\pm$ 1.67 | 3.37 $\pm$ 1.43 |
| H + Al (cmol$_c$ dm$^{-3}$) | 2.96 $\pm$ 0.57 | 2.96 $\pm$ 0.58 | 4.43 $\pm$ 1.04 | 5.25 $\pm$ 1.51 | 4.52 $\pm$ 0.74 |
| T-CEC (cmol$_c$ kg$^{-1}$) | 5.90 $\pm$ 0.66 | 4.70 $\pm$ 0.45 | 5.93 $\pm$ 0.80 | 7.51 $\pm$ 1.10 | 6.10 $\pm$ 1.13 |
| C:N | 4.52 $\pm$ 0.76 | 4.27 $\pm$ 0.66 | 3.53 $\pm$ 0.78 | 3.92 $\pm$ 0.66 | 4.75 $\pm$ 0.73 |
| Clay (dag kg$^{-1}$) | 19.0 $\pm$ 2.44 | 23.0 $\pm$ 5.78 | 23.0 $\pm$ 5.97 | 29.5 $\pm$ 5.25 | 29.0 $\pm$7.95 |
| Silt (dag kg$^{-1}$) | 16.2 $\pm$ 2.04 | 10.0 $\pm$ 2.15 | 17.0 $\pm$ 6.58 | 16.0 $\pm$ 11.47 | 21.2 $\pm$ 7.38 |
| Sand (dag kg$^{-1}$) | 68.0 $\pm$ 3.14 | 65.0 $\pm$ 5.14 | 57.0 $\pm$11.88 | 54.5 $\pm$ 15.75 | 49.8 $\pm$ 14.8 |

P1: active/slop; P2: active/slop; P3: active/flat; REG: natural regeneration, CON: control, SOM: soil organic matter, SMC: soil moisture content, T-CEC: potential cation exchange capacity. * Values significantly different ($p < 0.05$) in relation to restored areas and/or control site/pasture. ** Values significantly different ($p < 0.05$) in relation to control site/pasture.

### 3.2. Restoration Costs

In the case of the Atlantic Forest biome, the average costs of active restoration include the purchase of seedlings and labor (21.271 R$/ha) and the purchase of a fence (8.184 R$) [52]. Here, the scope of work includes the following activities: fencing the site, preparing the site (manual weeding, mowing) and planting trees. Cover fertilization costs were not considered. For natural regeneration, average costs include fence purchase (8.184 R$/ha) and labor (184 R$/ha). The costs of both strategies are summarized in Table 3.

**Table 3.** Comparison of the average costs of restoration strategies. The exchange rate during the implementation period (2006) was R$ 3.51/US$.

| Cost of Restoration | Active Restoration | Natural Regeneration |
|---|---|---|
| Purchase cost (seedlings and labour) | 21.271 R$/ha | 184 R$/ha |
| Purchase cost (fence) | 8.184 R$/ha | 8.184 R$/ha |
| Total costs of restoration (R$) | 29.455 R$/ha | 8.368 R$/ha |
| Total costs approx. (US$) | 8.392 US$/ha | 2.400 US$/ha |

## 4. Discussion

Poor pastureland management due to overgrazing and insufficient nutrient reposition can lead to the nutrient-rich topsoil depletion, reduction in soil fertility and biomass yields, followed by a loss of profit and income for the farmer. Consequently, unproductive pastures are abandoned and then exposed to erosion and leaching processes, which aggravates soil degradation [53–55]. Up to 18 million hectares of low-productivity pasture can be reforested in the Atlantic Forest, which is equivalent to the actual coverage of the remaining Atlantic Forest biome, without hindering the development of Brazilian agriculture [56]. Monitoring and understanding the belowground processes that occur during forest restoration is pivotal for sustainable land management and helpful to restoration planning and decision-making process. Here, analyzing short-term response (after six years) we found no significant differences in soil properties between restoration strategies, except for Na in P2 area and pH in P3 area (active plantation), compared to CON (abandoned pasture). Our results suggest that the recovery of soil properties on land formerly used for cattle grazing is a long-term process regardless of the restoration method. These findings corroborate previous studies conducted not only in the tropics but in other regions of the world, e.g., [57,58].

The highest content of Na in pasture soil may be associated with a difficult drainage caused by soil compaction, common in degraded pastures. Soils that show difficult drainage of the water favor the accumulation of Na in the superficial horizons [59]. Iron and manganese usually occur in low levels, and their amount depends on the material of origin and degree of soil weathering [60]. The ways in which these micronutrients are available to plants are related to soil properties such as organic matter, pH, and moisture [61–64]. In our study, soil from the hillside had significantly lower Fe and significantly higher Mn content in the comparison with the pasture located on the flat area. This may be explained by different soil types occurring in the study region. Soils from flat areas (P3, REG and CON) are classified as fulvic Neosols, typical for lowland [41], while Ultisols are predominant soils on the slope. Fulvic Neosols are soils derived from recent quaternary sediments, non-hydromorphic and with low pedogenetic development. Ultisols are highly weathered soils with accumulation of clay in the B horizon, typical of humid tropical environment [65,66]. Concentrations of Fe and Mn under active restoration in the slope might result from the composition of the parent material that gave rise to the Ultisols. Thus, we presume that independently of restoration strategy and landform, soil Fe and Mn content are related to soil type. In turn, soil pH from active/flat P3 area was significantly lower than active/slope P1 and P2. We did not observe difference between soil pH from natural regeneration and CON, in flat areas, in relation to P1 and P2. We suggest that this may result from tree species that were planted. Hong et al. [67] showed correlation between plants introduced for the forest regrowth and soil pH: when the tree species for forest establishing are selected based on initial soil pH value, afforestation may modify soil pH by increasing pH in acid soils or decreasing pH in alkaline soils. In general, forest soils are acidic with limited fertility [68,69]. This finding indicated another benefit of forest restoration: neutralizing of soil pH can potentially improve soil health and promote ecosystem productivity [67].

Soil moisture content was significantly lower in all restored areas when compared to the pasture area. There were no differences among treatments, indicating that the restoration strategy did not influence this process. Soil under forest is generally less humid than under other vegetation types such as shrubland or grassland since trees

reduce soil water content by evapotranspiration, infiltration, and canopy interception [70]. Evapotranspiration, the combined processes of direct evaporation and transpiration by plants, is influenced by changes in rooting characteristics, leaf area, stomatal response, and plant surface albedo [71]. Thus, evapotranspiration from forest is much higher than, for example, from pasture. In addition, infiltration capacity is much more expressive under forest areas due to the macroporosity, provided by death roots and animal canals [72]. Additionally, trees intercept rainfall water due to the leaf area size and roughness [72,73].

We found no significant difference between restoration strategies and CON, in terms of N and P concentrations. Regarding nitrogen, our results contrast with studies in secondary tropical forests in which N has a direct relationship with aboveground biomass [74–76] but are consistent with Holl and Zahawi [77] who report a similar result. Thus, even considering the well-known relationship between soil nutrient parameters and tree abundance, perhaps another factor here might be more relevant: the relation between nitrogen dynamics and the age of restoration. Nitrogen accumulation in the soil results from microbial decomposition of litter and roots originating from vegetation recovery [78,79]. Previous studies demonstrated that at the later restoration stage, forests have more fine roots than at earlier restoration stages [75,77]. Fine roots are easy to turnover [79]. As the number fine roots increase over vegetation succession, turnover of dead roots also increases and that may result in higher N concentration inputs into the soil [78,80]. The lack of a significant relationship between N and restored areas may have occurred since higher percentages of N content in soil are found in older restoration ages [76], and our study was conducted on areas restored for seven years. For example, Teixeira [81] reported the recovery of nutrient cycling to the levels of primary forests after 15 years of succession. Thus, longer restoration periods would be necessary for the results regarding soil N restoration to become more evident. In contrast to nitrogen, soil phosphorus is derived primarily from rock weathering and tropical soils are usually strongly weathered and have low total P content [82]. Moreover, low concentrations of P in soil during forest restoration might be caused by increased demands by plants due to biomass accumulation [83,84]. Different strategies of restoration in our study did not show significant difference of P in relation to CON. Here, the lowest average of P content was observed in CON and the highest average was observed in P1 area. We assume that here soil pH can be a key factor. Soil pH has been recognized an important factor influencing soil phosphorus availability [85]. The analyzed soils are acidic. In this case, inorganic P could react with Al and Fe ions, making it immobilized and non-available to plants [86].

Considering the dynamic of soil organic matter (SOM), we initially assumed that relatively short period of pasture-to-forest conversion might not be sufficient to restore initial carbon stock, which has been shown in previous studies [57,87]. However, the outcomes from other research that considers the accumulation rate of carbon in the soil are inconsistent. For example, Guo and Gifford [88] demonstrated that changing pasture lands to secondary forests results in soil carbon decrease. It is because in contrast with tree systems, pastures contain a higher number of fine roots that incorporate more carbon from their fast decomposition rate. Our results may also be explained by the presence of residual organic matter from the dominant grass in pasture. From the other hand, Carrasco-Carballido [58] and Nogueira [89] observed the increase of soil organic carbon content during both natural regeneration and planting after two and ten years, respectively. The authors suggested that the initial soil nutrient content and the use of leguminous tree species are crucial for fast soil organic matter recovery.

Finally, looking into socio-economic aspects, if we assume that the focus of restoration is the recovery of soil properties such as carbon stock and macronutrients content, natural regeneration, which is generally cheaper [90], would be better option. The active restoration costs for the current project were approximately three times higher than for natural regeneration. Additionally, the active strategy required soil correction and was more labor-intensive. Nevertheless, each restoration strategy has its implications and should be evaluated ad hoc [34,57,91,92]. Restoration projects should also consider neg-

ative aspects on improper land management, such as displace pre-existing agricultural activities (leakage), labor scarcity, or low tree survival [93–95]. Summing up, from the socio-economic point given early soil properties response, low-cost natural regeneration might be a preferred method to be proceeded on abandoned pastures in the study region.

Our study has limitations that should be mentioned. First, in case of field trials, environmental conditions are beyond the control of researchers. Second, access to some sampling sites and the soil sampling itself was restricted, which might have influenced the design of the experiment. Third, the results presented here are based on short-term studies. The authors are aware that such field studies are time consuming and require long-term monitoring to fully investigate soil changes during forest restoration. It should also be emphasized that preliminary soil data that we could use as reference was not available before the implementation of the forestry project. Since the restored areas were previously used for grazing cattle, we assumed that an abandoned pasture, used here as a control, would be the best reference system in terms of soil conditions. This weakens the experimental design in this work, because the lack of access to the original state of the restored areas from which soil samples were taken limits our ability to make conclusive statistical inferences with them. Our analyses aim strictly to guide future work in the field, providing hypotheses of early results to be tested under conditions that will allow for stronger experimental designs. At the end, we agree that our sample units cannot represent true replicates [96,97] given the distance between trial plots and restoration treatments. However, since all treatments were implemented in the same landscape, under similar environmental conditions such as forest cover and climate, we still believe that this experimental design could provide valuable information on tropical ecological restoration. The same approach has been used in several restoration initiatives due to the cost of implementing of true replicates, considering the wide environmental variation in the landscape (soil, past land use, climate, forest cover, etc.).

## 5. Conclusions

A range of criteria are considered to guide the decision of where and how to promote large-scale restoration efforts. These criteria include, inter alia, the potential of carbon sequestration, habitat availability, and restoration costs. Research on how different restoration strategies influence the recovery of soil properties and soil ecosystem services is scarce. This information is not only fundamental for long-term restoration success but also to monitor the below-ground changes that provide multiple ecosystem services. In the first six years after reforestation, we did not observe significant differences between the soils, except for the content of pH, Na, Fe, and Mn, which may be due to the individual properties of the evaluated soils. Our results suggest a slow recovery of the main physicochemical properties of soil in abandoned land that was previously used for grazing cattle, regardless of the restoration method. Considering the costs of both methods, natural regeneration would be preferable because it implies only mild soil preparation and fencing of the target areas to prevent excessive grazing by free ranging farm animals. Here, we highlight the importance of including ecosystem parameters as such as soil properties and land-use history in restoration projects. We conclude that the results obtained enrich the current body of knowledge on transformations occurring below the ground of the restored areas. Understanding these processes provides important practical and scientific information to promote more sustainable land management in the Brazilian Atlantic Forest. In future research, we recommend considering parameters such as infiltration and soil compaction as well as litter from the restored system and the relationship between vegetation cover and soil. We also suggest including soil analysis in other tropical regions of the world to maximize the success of ecological restoration efforts.

**Supplementary Materials:** The following are available online at https://www.mdpi.com/article/10.3390/land10080768/s1. Figure S1: Map of Brazilian Atlantic Forest (original and current vegetation cover); Figure S2: Variation of sodium content in each restored area with respect to control; Figure S3: Variation of iron and manganese content in each restored area with respect to control; Figure S4:

Variation of soil pH in each restored area with respect to control; Figure S5: Variation of soil moisture content in each restored area with respect to control; Figure S6: Variation of soil texture: clay, silt, and sand in each restored area with respect to control; Figure S7: Variation of soil macronutrients: total nitrogen, phosphorus, and potassium in each restored area with respect to control; Figure S8: Variation of soil organic matter in each restored area with respect to control; Figure S9: Variation of C:N ratio in soil in each restored area with respect to control; Table S1: Mean values and ± SD of soil properties (0–20 cm) of each restored area in Fazenda Dourada, Brazil; Table S2: Mean values and ± SD (n = 10) of soil properties (0–20 cm) from pasture/reference system (control) in Fazenda Dourada, Brazil; Table S3: Means values and ± SD of C:N ratio (0–20 cm) of each restored area and pasture/reference system (control) in Fazenda Dourada, Brazil; Table S4: List of the plant species in sites of active restoration (P1, P2, P3) and natural regeneration (REG) in Fazenda Dourada, Brazil, Word file: The quadratic propagation of standard deviations method description.

**Author Contributions:** Conceptualization, A.E.L. and J.B.B.S.; methodology, K.A.K., A.E.L. and M.S.M.; field data collection, K.A.K., M.S.M. and A.F.R.; writing—original draft preparation, K.A.K., A.E.L., M.S.M. and A.F.R.; statistical analysis, A.S.I.; writing—review and editing, A.E.L., J.B.B.S., A.F.R., V.D., C.C.J., A.A., I.A.B.P., E.L. and B.B.N.S.; visualization, K.A.K.; supervision, A.E.L.; project administration, A.E.L.; funding acquisition, A.E.L. All authors have read and agreed to the published version of the manuscript.

**Funding:** This research was funded by the International Climate Initiative (IKI), project title: Unlocking economic opportunities to scale Forest Landscape Restoration in Brazil, project number 17_III_089; Global Environment Facility (GEF), project title: Realizing the biodiversity conservation potential of private lands in Brazil, project ID: 1 9413, Norwegian Agency for Development Cooperation (Norad), project title: Land Neutral Agricultural Expansion and Ecological Restoration in Brazil, Agreement Number: QZA–0461BRA–13/0002; Newton Fund (NAF\R2\180676), Fundação Carlos Chagas Filho de Amparo à Pesquisa do Estado do Rio de Janeiro (Faperj) (E-26/202.680/2018) and Conselho Nacional de Desenvolvimento Científico e Tecnológico (CNPq) (308536/2018-5) for the project "Sustaining the land from the ground up: developing soil carbon and soil ecosystem services valuation frameworks for tropical soils". This study was also partially financed by the Coordenação de Aperfeiçoamento de Pessoal de Nível Superior (CAPES)—Finance Code 00.

**Institutional Review Board Statement:** Not applicable.

**Informed Consent Statement:** Not applicable.

**Data Availability Statement:** All data generated and analyzed is included in the manuscript.

**Acknowledgments:** We thank Associação Mico-Leão Dourado (AMLD) for enabling field study and technical support. J.B.B.S. is supported by CNPq grants (PQ-2-Productividade em Pesquisa). We would like to thank G. Galvão for all help and support in field data collection. We also thank the anonymous reviewers for their constructive comments and suggestions.

**Conflicts of Interest:** The authors declare no conflict of interest.

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
