# Peer review of "Early Response of Soil Properties under Different Restoration Strategies in Tropical Hotspot"

_land, doi:10.3390/land10080768_

Round 1

Reviewer 1 Report

The paper aims at understanding the changes of tropical soil properties in early restoration stage of the Brazilian Atlantic Forest.

Introduction: The study is clearly motivated but the authors should provide numerical examples from the papers they cite to better support their statements

Materials and Methods

Study site: This part is not clear because I understand that the authors describe a network of protected areas near the place where they made their sampling. Two sentences are not logically connected to the rest ("Fazenda dourada..."; The preservation area...") . The characters in the maps of figure 1 are too small when printed at 100% size.

Experimental design: This part is weak or even faulty. Indeed, as the design is presented, there lacks repetitions of treatment for  control and natural regeneration. But, perhaps the various areas are sufficiently large and the three blocks in each treatment could be considered as the experimental units? In any way, this is a serious problem which will limit the soundness of any conclusions.

Characteristic of the aboveground vegetation: the authors should provide a summary table for clarity.

Soil sampling and analyses: For organic carbon, NaCr2O7 in H2SO4 is not used for extraction of organic C but for oxidation .

Statistical analyses: A brief explanation of the principles of the method 'quadratic propagation of standard deviations' should be given because as it is uncommon.

Results

Soil properties: In Table 1, there lack indications of the series which are different. Perhaps the granulometric data should be given in the Materials and methods section as initial differences between the areas, or do they potentially change during regeneration?

Socio-economic aspects: the paragraph should be given mostly  in Materials and Methods section. Table 2, whose object is socio-economic aspects, gives data for soil properties and biodiversity without translation into ecosystem services; labour should be translated in R$ or better in an international currency, as the other estimations

Discussion

Due to the weak nature of the design, I feel the discussion mostly as speculation to explain the differences between the treatment.

Paragraph at line 334 should be in the Result section

l245 it could be added that thos 18 million hectares are as much as the actual cover of remaining BAF.

Reviewer 2 Report

The work is interesting and well structured. It falls within the framework of applied research, aimed at making research results immediately operational. The topic addressed is of great interest, especially under the increasing pressure of climate change. The purpose is to compare different restoration strategies using as criteria physicochemical characteristics that are significant of soil "quality" and that are a measure of the functions that soil can perform in favor of human well-being (agricultural productivity, resistance to erosion or disruption, carbon storage, and many other ecosystem services). 

The paper is well-written and the diverse background of the authors suggests that the structure was deliberately geared more toward the applicability of the results than an in-depth description of the methodology and methods of analysis used. This approach, by the way immediately declared by the authors, doesn't take away at all the merit of the work that, on the contrary, results easy to read and of immediate comprehension for a wider audience.

To me, the paper is worthy of publication. The only advice is to spend a few words on the replicability of the same methodology (choice of parameters considered, choice of species considered, etc...) in other contexts or in other regions of the world.

Round 2

Reviewer 1 Report

This revised manuscript is substantially improved. I also agree with author’s reply but I regret that they did no pass several of their arguments into their text (see below). Unfortunately, the  significance of the work remains low because of the short-term of the study but also of the weak design. This one is not under the responsibility of the authors and they have to explain that there is no alternative option to get information on their topic. I suggest below a reference dealing with this problem.

Particular comments

Abstract:

L28: change ‘provides’ to ‘should provide’

L27 change ‘the impact of different…” to ‘the impact of those different…’

L32 change ‘cattle grazing. An abandoned pasture…’ to ‘cattle grazing and compared to an abandoned pasture as a reference system.’

L36 add the cost

Materials and Methods

There are several problems with the upper et the lower case characters and typing

L137 ‘in on’ ?

L141 ‘soil preparation’ should be explained

Fig. 1 provide also a map (with scale) with plot positions and if possible, block positions

L161 how many blocks in CON?

Use the abbreviations throughout the text. Particularly, the control plots (CON) are called ‘control plots’, ‘pastures’, ‘control site’ below

L212 my comment in the first review about the position in Materials and methods or results did only concern plant composition, not the costs. Provide rather a table with the costs

Results

L135 suppress first sentence and add ‘ (Table 2)’ at the end of sentence 2

L265 the comments and the table on the costs are lacking

Discussion

L278 add ‘ (after 6 years)’ after ‘short-term response’ and add

L280 add ‘ (active plantation) compared to CON (abandoned pasture) ’ after ‘…for Na in P2 area and pH in P3 area’

After the previous sentence, I think I would be very valuable to be more critical with the design and thus the statistical quality of the samples, but also to explain why. The ‘why’ is clearly exposed in the introduction of your reply letter. Arguments for nevertheless using samples collected in a weak experimental design could be found in Davies & Gray, 2015. Don’t let spurious accusations of pseudoreplication limit our ability to learn from natural experiments (and other messy kinds of ecological monitoring), Ecology and Evolution, 5, 5295-5304, doi: 10.1002/ece3.1782)

L315 suppress ‘consumption’

Conclusions

L382 Maybe complete the sentence? ‘Considering the costs of both methods, natural regeneration would be preferable because it implies only mild soil preparation and fencing of the target areas to prevent excessive grazing by free ranging farm animals’
